# Liver Transudate, a Potential Alternative to Detect Anti-Hepatitis E Virus Antibodies in Pigs and Wild Boars (*Sus scrofa*)

**DOI:** 10.3390/microorganisms8030450

**Published:** 2020-03-23

**Authors:** Alejandro Navarro, Carmen Bárcena, Pilar Pozo, Alberto Díez-Guerrier, Irene Martínez, Coral Polo, Clara Duque, David Rodríguez-Lázaro, Joaquín Goyache, Nerea García

**Affiliations:** 1VISAVET Health Surveillance Center, Complutense University of Madrid, 28040 Madrid, Spain; angomez@ucm.es (A.N.); cbarcena@ucm.es (C.B.); ppozo@ucm.es (P.P.); irmartin@visavet.ucm.es (I.M.); coral.polo@ucm.es (C.P.); mauretaniagarden@gmail.com (C.D.); jgoyache@visavet.ucm.es (J.G.); 2MAEVA SERVET, S.L., Alameda del Valle, 28749 Madrid, Spain; adsmaeva@hotmail.com; 3Microbiology Division, Faculty of Sciences, Universidad de Burgos, Plaza Misael Bañuelos s/n, 09001 Burgos, Spain; 4Department of Animal Health, Faculty of Veterinary Medicine, Complutense University of Madrid, 28040 Madrid, Spain

**Keywords:** hepatitis E virus, pig, wild boar, liver transudate, diagnosis, serology, ELISA, PCR

## Abstract

In recent years, cases of hepatitis E virus (HEV) infection have increased in Europe in association with the consumption of contaminated food, mainly from pork products but also from wild boars. The animal’s serum is usually tested for the presence of anti-HEV antibodies and viral RNA but, in many cases such as during hunting, an adequate serum sample cannot be obtained. In the present study, liver transudate was evaluated as an alternative matrix to serum for HEV detection. A total of 125 sera and liver transudates were tested by enzyme-linked immunosorbent assay at different dilutions (1:2, 1:10, 1:20), while 58 samples of serum and liver transudate were checked for the presence of HEV RNA by RT-qPCR. Anti- HEV antibodies were detected by ELISA in 68.0% of the serum samples, and in 61.6% of the undiluted transudate, and in 70.4%, 56.8%, and 44.8% of 1:2, 1:10, or 1:20 diluted transudate, respectively. The best results were obtained for the liver transudate at 1:10 dilution, based on the Kappa statistic (0.630) and intraclass correlation coefficient (0.841). HEV RNA was detected by RT-qPCR in 22.4% of the serum samples and 6.9% of the transudate samples, all samples used for RT-qPCR were positive by ELISA. Our results indicate that liver transudate may be an alternative matrix to serum for the detection of anti-HEV antibodies.

## 1. Introduction

Hepatitis E virus is the main cause of acute hepatitis worldwide, causing more than 20 million infections and more than 60,000 deaths per year [1]. HEV was first characterized in 1955 from an outbreak in New Delhi, India, when it was initially named non-A/non-B hepatitis. In 1983, the virus was named HEV to reflect its tendency to cause epidemics of enteric infections [2]. Hepatitis E infections in Europe have historically been caused by travelers returning from endemic countries, but in the last decade, autochthonous cases have greatly increased and have been associated with zoonotic transmission, mainly through the consumption of poorly cooked pork meat or liver-derived meat products [3,4,5].

HEV belongs to the Orthohepevirus genus and the *Hepeviridae* family, and it is classified into eight genotypes according to the genetic sequence [6]. Genotypes 1 and 2 infect humans exclusively, are endemic in developing countries, and appear in epidemic forms [1]. These genotypes are mainly transmitted by the fecal-oral route, through consumption of contaminated water. Genotypes 3 and 4 are zoonotic, affecting humans and several animal species around the world. They are responsible for autochthonous and sporadic infections, and they are the main cause of hepatitis E infections in industrialized countries [5,6]. Genotype 3 has been detected in a wide variety of animals such as horses, mongooses, rats, rabbits, cows, dogs, cats, chickens, and red deer. Genotype 4, in contrast, affects pigs, wild boars, and donkeys nearly exclusively, and it is restricted to Asia [7,8,9,10,11,12]. Transmission of these genotypes occurs mainly through the consumption of raw or undercooked meat or liver products from its main host, the pig, but also from wild boars and deer [3,13,14]. Human populations that maintain close contact with animals show higher hepatitis E seroprevalence [15,16,17]. Non-animal transmission routes have also been described for these genotypes, including organ transplantation, blood transfusion [4,18,19,20,21,22,23], and vertical transmission [24,25]. More recently, genotypes 5 and 6 have been described in wild boars and genotypes 7 and 8 in camelids in Asia [26,27,28,29].

In a majority of cases, HEV genotype 3 causes asymptomatic disease. In some individuals, such as immunosuppressed patients, transplant recipients, or those with previous liver disease, a viral infection might result in chronic hepatitis [30,31]. Infection of HEV genotype 3 can also cause extrahepatic symptoms, including neurological symptoms, hematological dysfunction, and decreased glomerular filtration rate [32,33,34].

Genotype 3 was first identified from a pig in the USA in 1997 when it was shown to be related both antigenically and genetically to human HEV [35]. Since then, pigs have been described as the main reservoir for this genotype [5]. HEV prevalence in swine populations varies widely with study and country [36,37]: one review found seroprevalences of 30–93% at the farm level (overall HEV prevalence of 10–100%) and 8–93% at the population level (overall HEV prevalence of 1–89%). These wide ranges likely reflect variation in several factors such as farming practices, biosecurity measures, or the number of pigs, and sows on the farm [38]. 

Another relevant HEV animal reservoir is wild boars, and increasing cases of human HEV infection due to the consumption of infected wild boar meat have been reported in Asia and Europe in recent years [3,39]. Nevertheless, the reported prevalence in wild boar is mostly lower than that in pigs [3,9,40,41,42]. Wild boars may play an important role in the epidemiology of HEV in swine populations since transmission from wild boars to domestic pigs has been experimentally demonstrated [43].

The enzyme-linked immunosorbent assay (ELISA) technique is the main diagnostic method to perform HEV surveillance studies, and many commercial and ’in-house’ kits have been described [44,45,46]. Serum samples are typically used to assay anti-HEV antibodies, although meat juice and body cavity transudate have also been used in a few studies [47,48]. To detect the presence of HEV RNA, several reverse transcriptase-polymerase chain reactions (RT-PCR) techniques have been described, with the method proposed by Jothikumar et al. in 2006, one of the most frequently used because of its high specificity and sensitivity [49,50]. Samples most frequently used for RT-PCR detection are feces, liver, serum, muscle, semen, and food products (e.g., sausage, meat). Feces and liver are the matrices of choice since the virus can be detected in serum only for a short time [51]. For field prevalence hepatitis E studies, the use of RT-qPCR cannot always be affordable (technically and economically) due to the number of samples to be tested. In addition, even for seroprevalence studies, optimal serum samples are sometimes difficult to obtain and even not possible, particularly in the case of wild boar hunts, where a long period of time can pass between animal death and sampling. Therefore, the aim of the present study was to evaluate the use of liver transudate as an alternative matrix for the detection of HEV-antibodies using a commercial ELISA kit, as well as for the detection of HEV RNA by RT-qPCR.

## 2. Materials and Methods

### 2.1. Animal Samples

A total of 125 paired serum and liver transudate samples from 44 white pigs, 46 Iberian pigs, and 35 wild boars were included in the present study. Blood samples and liver samples (25 g) were collected at the different slaughterhouses in the case of pigs from the Spanish provinces of Cuenca, Pontevedra, Barcelona, Zaragoza, Burgos, Málaga, Córdoba, Girona, Toledo, Murcia, and Badajoz. Wild boar samples were recovered in the field after approximately 6 hours after the animal’s death by hunting, in the province of Madrid, Spain. All samples were taken by authorized veterinary personnel and transported to the laboratory in refrigerated containers. Once in the laboratory, blood was preserved at 4 °C until the next day, when serum was obtained and stored at −40 °C for the ELISA and at −80 °C for the RT-qPCR. Liver samples were stored at −80 °C. In order to obtain the transudate, liver samples were frozen and thawed once at room temperature. The resulting liquid was collected between four and eight hours later and then stored at −40 °C for ELISA and at −80 °C for RT-qPCR. All animal experiments in this study were conducted according to Spanish regulations (RD 53/2013) and European regulations (EU Directive 2010/63/EU).

### 2.2. Detection of Anti-HEV Antibodies by ELISA

All sera and liver transudate samples were analyzed for the presence of anti-HEV antibodies using the commercial ELISA kit ID Screen Hepatitis E Indirect Multi-species (IDvet, Montepellier, France) according to the manufacturer’s specifications, with a cut-off value set at 70 (S/P%). This is a duplicate-well test in which even-numbered wells are coated with a recombinant antigen from the capsid of HEV genotype 3, and odd-numbered wells are uncoated. Serum was diluted at 1:20 in the dilution buffer provided by the kit, while transudate was tested using the original undiluted sample as well as the dilutions 1:2, 1:10, and 1:20 in the dilution buffer. All plates were read at 450 nm with a Zenyth 3100 microplate multimode detector (Anthos Labtec Instruments GmbH, Salzburg, Austria).

### 2.3. Detection of HEV RNA by RT-qPCR

RT-qPCRs was performed on 58 paired serum and liver transudate samples from subsets of the animals (35 white pigs, 5 Iberian pigs, and 18 wild boars). These subsets were selected based on the amount of sample available. HEV RNA was obtained using the QIAmp Viral Mini Kit (QIAGEN, Hilden, Germany) according to the manufacturer´s instructions. The obtained product was stored at −80 °C until RT-qPCR was performed. HEV viral genome was detected using a previously described RT-qPCR assay [52] with minor modifications and a CFX96™ Real-Time Thermocycler System (Bio-Rad Laboratories, Munich, Germany). RT-qPCR was performed using the QuantiFast Pathogen RT-PCR + IC kit (QIAGEN) in a 20-µL reaction volume containing 10 µL of RNA. Reverse transcription was performed at 50 °C for 30 min, followed by denaturation at 95 °C for 5 min and 45 cycles of 95 °C for 10 s, 60 °C for 20 s, and 72 °C for 15 s. Samples showing PCR inhibition were diluted 1:10 in RNase free water.

### 2.4. Statistical Analysis

Animal type and the number of frozen-thaw cycles were tested in univariable logistic regression models using HEV positivity as the outcome variable. Risk factors that were significant in the univariable model with a generous *p*-value range < 0.2 were considered for inclusion in a multivariable model. The agreement between the qualitative results obtained in serum and liver transudate at different dilutions was measured using the Kappa statistic (κ). The corresponding optical densities (ODs) were compared using the intraclass correlation coefficient (ICC). All analyses were performed using IBM SPSS Statistics 25.0 (IBM, Armonk, NY, USA). When appropriate, results were reported together with their 95% confidence intervals (CIs). Sensitivity, specificity, and the area under the ROC curve (AUC) were calculated with the “pROC” [53] and “ROCR” [54] packages of R 3.3.3 [55].

## 3. Results

### 3.1. Serological Results

A total of 85 of 125 serum samples analyzed by ELISA were positive, which indicated an overall seroprevalence of 68.0% (95%CI 59.3–75.5). The prevalence was 77.2% (95%CI 63.0–87.1%) in white pigs (34/44), 63.0% (95%CI 48.6–75.4%) in Iberian pigs (29/46), and 62.8% (95%CI 46.3–76.8%) in wild boars (22/35). Of the 125 liver transudate samples, 77 tested positive when undiluted (61.6%, 95%CI 52.9–69.7%), 88 when diluted 1:2 (70.4%, 95%CI 61.9–77.7%), 71 when diluted 1:10 (56.8%, 95%CI 48.0–67.5%), and 56 when diluted 1:20 (44.8%, 95%CI 36.4–53.5%) (Table 1).

The prevalence was different among the different animal species (Table 1). The difference in HEV prevalence between transudate and serum assays ranged from 2 to 23.2 percentage points, depending on the dilution (Table 1). Specificity, sensitivity, positive predictive value (PPV), and negative predictive value (NPV) for the HEV ELISA are displayed in Table 2, together with the area under the curve (AUC) using confidence intervals of 95%, and a cut-off value of 70%. The highest overall specificity was obtained with the 1:20 transudate dilution (95.0%, 95% CI 87.5–100) followed by the 1:10 transudate dilution (90.0%, 95% CI 80.0–97.5). The highest sensitivity was obtained with the 1:2 transudate dilution (85.9%, 95% CI 77.6–95.9), followed by the undiluted transudate (82.3%, 95% CI 74.1–90.6). The best PPV corresponded to the 1:20 transudate dilution (96.5%, 95% CI 91.1–100), followed by the 1:10 transudate dilution (94.4%, 95% CI 89.2–98.6). The best NPV was obtained with undiluted transudate (69.1%, 95% CI 59.3-79.6), followed by 1:2 transudate dilution (67.6%, 95% CI 55.3–81.3) and 1:10 transudate dilution (66.7%, 95% 58.2–77.1). Finally, similar AUC values were obtained across the samples, with pairwise differences ranging from only 0.6 to 1.2 percentage points (Table 2).

In the univariate logistic regression analysis, no association was found between animal species and the results obtained in serum or transudate (*p* > 0.05). The 1:10 dilution of the liver transudate showed the best agreement with serum, based on the kappa coefficient (κ = 0.630) (Table 1). ICCs for the comparison of ODs between serum and transudate dilutions were as follows: undiluted, 0.820; diluted 1:2, 0.803; diluted 1:10, 0.841; and diluted 1:20, 0.787 (Figure 1). The ROC curve, sensitivity, specificity, and the AUC for the undiluted and diluted liver transudates are shown in Table 2. 

### 3.2. RT-qPCR Detection

A total of 58 paired serum and liver transudate samples were analyzed by RT-qPCR (Table 3). In total, 13 serum (22.4%, 95%CI 13.5–34.6%) and four liver transudate samples (6.90%, 95%CI 2.71–16.4%) were positive. Of the four positive liver transudates, three of the paired serum samples were also positive, with threshold cycle values (Ct) of 33.27, 36.90, and 37.28 for the serum and 27.91, 24.98, and 32.78 for the transudate samples (Table 4). The ELISA reactions for these three animals were also positive for serum and all liver transudate dilutions. Interestingly, the animal with an RT-qPCR positive transudate sample but RT-qPCR negative for serum-tested ELISA positive for both serum and all liver transudate dilutions. Of the 13 RT-qPCR-positive serum samples, eight were from the white pig while the remaining five were from wild boars. All four of the RT-qPCR-positive transudate samples were obtained from wild boars. Generally, Ct values were lower in transudate (27–40) than in serum (28–42).

All 17 RT-qPCR-positive serum and transudate samples were also positive by ELISA, except for three samples that were positive by RT-qPCR but negative by ELISA in the 1:20 liver transudate dilution. A total of 43 of 45 sera samples were RT-qPCR-negative and positive by ELISA. Of the 54 RT-qPCR-negative transudate samples, 54 undiluted samples, 54 dilutions of 1:2, 51 dilutions of 1:10, and 41 dilutions of 1:20 were positive by ELISA.

## 4. Discussion

The present study is the first to demonstrate that liver transudate may be a good alternative matrix for the detection of anti-HEV antibodies in pigs and wild boars, since our results suggest a good agreement between the qualitative results and the optical densities of both matrices in the ELISA technique, especially with the 1:10 liver transudate dilution.

Although other alternatives to serum have been tested for the detection of anti-HEV antibodies, they are scarce, and the results have not always been compared with the serum results. A previous study detected anti-HEV antibodies in body cavity transudate obtained from raccoons, raccoon dogs, cats, and dogs [47]. Meat juice has also been used in some studies to determine HEV seroprevalence in hunted wild boars and slaughtered pigs with satisfactory results [48,56]. Similarly, other reports also demonstrated the utility of meat juice to determine the prevalence of different agents such as *Salmonella*, *Trichinella*, influenza virus or *Toxoplasma* [57,58]. Meat juice, like other body fluids, contains fewer antibodies against HEV, *Toxoplasma gondii*, and other pathogens than serum [56,57,58,59], so the dilution factor in serological assays should be adjusted accordingly. Therefore, liver transudate was tested undiluted and at different dilutions to determine this factor. 

In this study, anti-HEV antibodies were successfully detected in the liver transudate of pigs and wild boars, and their detection rate showed an elevated agreement with their paired serum samples. The specificity and sensitivity obtained with liver transudate were high, which indicates that it could be a good alternative to serum, which is in concordance with other studies made in another matrix [56]. Both serum and liver transudates at all dilutions presented a range of very similar OD values. In general, seroprevalence in serum was slightly higher than the one obtained with the liver transudate, which could be explained by the presence of inhibitors of the ELISA in the latter type of samples. Meat or liver exudates are tricky analytical templates and, as other many food items, they contain components, such as fat or haemo group, that can produce a matrix effect in the ELISA reaction [59,60], interfering with the ELISA by inhibiting antigen-antibody binding, reacting with epitopes, or having interfering enzymatic activity [61]. These results required optimizing the dilution factor, as reported for other matrices such as meat juice [62]; we found 1:10 to be the best dilution for detecting anti-HEV antibodies using a commercial indirect ELISA test. Although this dilution did not always give the best individual values of specificity, sensitivity, PPV, or NPV, it gave overall the best results. In addition, the 1:10 transudate dilution showed the best concordance of serum results with qualitative and optical densities in ELISA results.

Only 13 serum and four liver transudate samples out of 58 were RT-qPCR positive, and only three samples were RT-qPCR-positive for both matrices from the same animal. The presence of antibodies in serum and liver transudate was substantially higher than the presence of the virus detected by RT-qPCR in both matrices, in agreement with a previous study that found high seroprevalence 73.3% but the prevalence of only 15.6% in liver respectively based on PCR [52]. In the present study, all RT-qPCR-positive samples in both matrices were also positive by ELISA. Most serum samples (43/45) and all transudates that were negative by RT-qPCR were positive by ELISA. This may be because viremia lasts for a very short time in circulating blood, which makes it quite difficult to detect viral RNA in serum [51] while circulating antibodies persist.

## 5. Conclusions

The obtained results suggest that liver transudate, used at an appropriate dilution (1:10), may be a good alternative matrix to serum for detecting anti-HEV antibodies by ELISA in pigs and wild boars. This matrix may be useful in postmortem analyses for diagnosis, surveillance, or research in field studies, particularly when obtaining blood is impractical. Although HEV could be detected by RT-qPCR in the liver transudate, further studies with more samples are needed to obtain any conclusion.

## Figures and Tables

**Figure 1 microorganisms-08-00450-f001:**
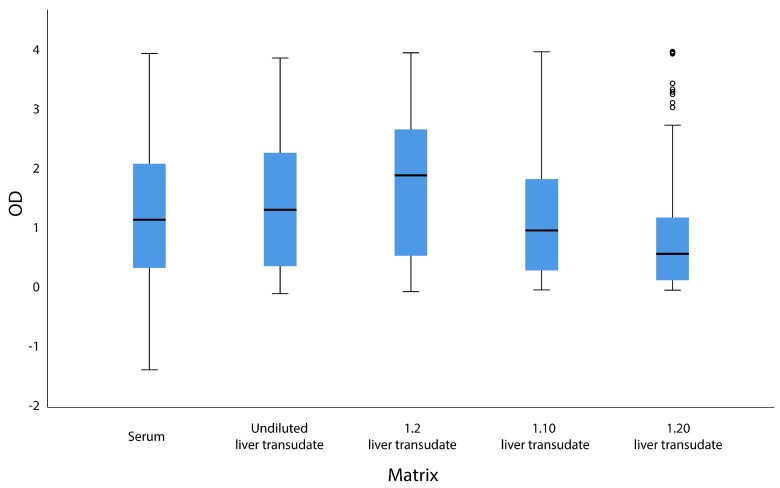
Box plot diagram showing the optical densities (OD) for serum and undiluted or diluted liver transudate (1:2, 1:10, or 1:20).

**Table 1 microorganisms-08-00450-t001:** Numbers of serum and liver transudate samples positive for antibodies against hepatitis E virus by ELISA.

Species (*n*)	Positive Serum Samples, *n* (%)	Positive Undiluted Liver Transudate Samples, *n* (%)	Positive Transudate 1:2 (%)	Positive Transudate 1:10 (%)	Positive Transudate 1:20 (%)
White pig (44)	34 (77.2)	37 (84.0)	38 (86.3)	34 (77.2)	28 (63.6)
Iberian pig (46)	29 (63.0)	20 (43.4)	29 (63.0)	17 (36.9)	11 (23.9)
Wild boar (35)	22 (62.8)	20 (57.1)	21 (60.0)	20 (57.1)	17 (48.5)
All (125)	85 (68.0)	77 (61.6)	88 (70.4)	71 (56.8)	56 (44.8)
κ		0.616	0.494	0.630	0.491

**Table 2 microorganisms-08-00450-t002:** Agreement between positive and negative results obtained by ELISA in transudate compared with serum, specificity, sensibility, Positive Predictive Value (PPV); Negative Predictive Value (NPV) and Area Under the ROC curve AUC) for HEV antibody assays conducted on different dilutions of the liver transudate.

Transudate Dilution	Samples *n* (%)		Specificity, % (95% IC)	Sensitivity, % (95% IC)	PPV, % (95% IC)	NPV % (95% IC)	AUC, % (95% IC)
Undiluted	Positive	70 (82.3)	82.5 (70.0–92.5)	82.3 (74.1–90.6)	91.1 (85.5–96.0)	69.1 (59.3–79.6)	89.7 (83.9–95.4)
	Negative	33 (82.5)					
1:2	Positive	73 (85.8)	62.5 (47.5–77.5)	85.8 (77.6–92.9)	82.8 (77.1–88.9)	67.6 (55.3–81.3)	88.5 (82.6–94.4)
	Negative	25 (62.5)					
1:10	Positive	67 (78.8)	90.0 (80.0–97.5)	78.8 (69.4–87.1)	94.4 (89.2–98.6)	66.7 (58.2–77.1)	88.7 (82.2–95.2)
	Negative	36 (90.0)					
1:20	Positive	54 (63.5)	95.0 (87.5–100)	63.5 (54.1–72.9)	96.5 (91.1–100)	55.1 (48.1–62.9)	89.1 (83.3–94.9)
	Negative	38 (95.0)					

**Table 3 microorganisms-08-00450-t003:** Numbers of serum and liver transudate samples positive for hepatitis E virus by polymerase chain reaction (PCR).

Species (*n*)	Serum Samples, *n* (%)	Liver Transudate Samples, *n* (%)
White pig (35)	8 (22.8)	0
Iberian pig (5)	0	0
Wild boar (18)	5 (27.7)	4 (22.2)
All (58)	13 (22.4)	4 (6.9)

**Table 4 microorganisms-08-00450-t004:** Threshold cycle (Ct) of positive samples obtained by the RT-qPCR.

Sample ID	Serum Samples, Ct	Liver Transudate Samples, Ct
**1**	33,27	27.91
**2**	41,06	N/A
**3**	28,04	N/A
**4**	37,63	N/A
**5**	36,90	24.98
**6**	41,08	N/A
**7**	41,96	N/A
**8**	38,96	N/A
**9**	40,57	N/A
**10**	42,21	N/A
**11**	36,90	N/A
**12**	42,62	N/A
**13**	37,28	32,78
**14**	N/A	40

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
