# Peer review of "Liver Transudate, a Potential Alternative to Detect Anti-Hepatitis E Virus Antibodies in Pigs and Wild Boars (Sus scrofa)"

_microorganisms, 2020, doi:10.3390/microorganisms8030450_

Round 1

Reviewer 1 Report

This manuscript describes the assessment of liver transudate vs serum for the diagnostic detection of HEV antibodies. A reasonable alternative but some further detail is required. It would also be helpful if the authors suggest this is useful for antibody detection and why this may be the case, e.g., a situation this information would be useful might be purely for prevalence information or would it be considered a good option for viral detection in the case of transmission?

Introduction:

line 16-17 just to comment has this been formally proven?

line 25 on, the abstract needs some amending to indicate that all PCR were positive by ELISA.

line 49 G4 found elsewhere now so define please.

Methods:

line 97, in methods please provide more detail on how the transudate is obtained, no detail is provided. What is the sensitivity and specificity of the commercial assays that you chose to use? Are they known to adequately detect HEV in porcine species?

For the PCR, 2.3, how was the RNA integrity determined? How were the samples controlled for the presence of viral DNA?

The authors mentioned that the samples showing inhibition were diluted, how many neat samples was this? How can you control for false negatives and compare dilutions?

Results:

Figure 1 does not seem necessary given the information on table 1, I would suggest that this can be removed.

Discussion and Conclusion:

This needs more comment on the validity of the results based on the assays used.

Line 201 - states meat juice contains fewer antibodies, so why dilute out transudate, do you expect this to be less too? Please clarify.

Line 204 - relevant agreement - what does this mean?

Line 206 - why did you dilute the transudate; were levels too high/low? What is the explanation for this as sensitivity and specificity are not discussed.

Why were Cts lower in transudate vs serum? Please comment. Is this a better source than serum? Is the virus persistent here? No comment here on how this compares to information on PCR in the literature;

line 215 ref 49 is this correct?

No comment made on wider range of OD's in serum vs transudate.

Reviewer 2 Report

Microorganisms 736334

Liver transudate, a potential alternative to serum for diagnosis of hepatitis E virus infection in pigs and wild boars (Sus scrofa)

The purpose of this study was to evaluate the use of liver transudate as an alternative matrix to serum to detect the presence of antibodies against HEV in pigs and wild boar. This study is well done, well written, very clear and relevant. Some minor corrections will have to be made.

Title: The authors talk about infection but it is more a question of assessing seroprevalence. This clarification should be made.

Line 23 : HEV was not detected by ELISA, the antibodies anti-HEV were detected.

Line 41: The taxonomy needs to be checked because it's been changed and the genus is now Orthohepevirus.

Line 71: References should be added

Line 140: The samples were not positive HEV virus by ELISA, only the antibodies against HEV were detected by ELISA.

Reviewer 3 Report

The introduction section could be improved by providing stronger rationale for the use of liver transudates. What are the advantages and disadvantages of using liver transudates compared to serum and liver? Since liver transudates cannot be obtained from live animals, and the data presented showed that at best the results are comparable to serum for ELISA, except for screening of game meat, I do not see any advantage of using liver transudates.

How do the qRT-PCR data from liver transudates compare to liver matrix?

There is no clear definition of liver transudate and the authors did not provide a clear description of the procedures for preparing liver transudate.

Please provide a separate table for the obtained Ct values.

The discussion could be improved by explanation of potential ELISA inhibitors in transudates.

L41 please italicize hepeviridae

L42 please change to genetic sequence

L59 HEV infection, in these instances, might result in chronic infection, it is not definitive though.

L91 where were the samples collected? Please provide the geographical location for the samples.

What are the error bars in figure 1?

L211 please rephrase.

Round 2

Reviewer 3 Report

It would much more helpful if the authors provided the full response instead of referring to changes made in the revised manuscript.

L103, Remove the before ELISA

L218, Change what to which

L219-220, Please rephrase

No actual discussion on potential ELISA inhibitors. What is their nature? Have they been described before? where?
